# Multispectral Depth-Resolved Fluorescence Lifetime Spectroscopy Using SPAD Array Detectors and Fiber Probes

**DOI:** 10.3390/s19122678

**Published:** 2019-06-13

**Authors:** João L. Lagarto, Caterina Credi, Federica Villa, Simone Tisa, Franco Zappa, Vladislav Shcheslavskiy, Francesco Saverio Pavone, Riccardo Cicchi

**Affiliations:** 1National Institute of Optics, National Research Council (INO-CNR), Via Nello Carrara 1, 50019 Sesto Fiorentino, Italy; pavone@lens.unifi.it (F.S.P.); rcicchi@lens.unifi.it (R.C.); 2European Laboratory for Non-linear Spectroscopy (LENS), Via Nello Carrara 1, 50019 Sesto Fiorentino, Italy; credi@lens.unifi.it; 3Department of Information Engineering (DINFO), University of Florence, Via di S. Marta 3, 50139 Firenze, Italy; 4Dipartimento di Elettronica, Informazione e Bioingegneria (DEIB), Politecnico di Milano, 20133 Milan, Italy; federica.villa@polimi.it (F.V.); franco.zappa@polimi.it (F.Z.); 5Micro Photon Device SRL, Via Waltraud Gebert Deeg 3g, I-39100 Bolzano, Italy; stisa@micro-photon-devices.com; 6Becker & Hickl GmbH, Nunsdorfer Ring 7-9, 12277 Berlin, Germany; vis@becker-hickl.de; 7Department of Physics, University of Florence, Via G. Sansone 1, 50019 Sesto Fiorentino, Italy

**Keywords:** SPAD, CMOS, fluorescence spectroscopy, fluorescence lifetime, depth-resolved fluorescence, tissue diagnosis, fiber optics

## Abstract

Single Photon Avalanche Diode (SPAD) arrays are increasingly exploited and have demonstrated potential in biochemical and biomedical research, both for imaging and single-point spectroscopy applications. In this study, we explore the application of SPADs together with fiber-optic-based delivery and collection geometry to realize fast and simultaneous single-point time-, spectral-, and depth-resolved fluorescence measurements at 375 nm excitation light. Spectral information is encoded across the columns of the array through grating-based dispersion, while depth information is encoded across the rows thanks to a linear arrangement of probe collecting fibers. The initial characterization and validation were realized against layered fluorescent agarose-based phantoms. To verify the practicality and feasibility of this approach in biological specimens, we measured the fluorescence signature of formalin-fixed rabbit aorta samples derived from an animal model of atherosclerosis. The initial results demonstrate that this detection configuration can report fluorescence spectral and lifetime contrast originating at different depths within the specimens. We believe that our optical scheme, based on SPAD array detectors and fiber-optic probes, constitute a powerful and versatile approach for the deployment of multidimensional fluorescence spectroscopy in clinical applications where information from deeper tissue layers is important for diagnosis.

## 1. Introduction

Autofluorescence spectroscopy explores the optical properties of endogenous molecules, such as collagens, reduced nicotinamide adenine dinucleotide (phosphate) (NAD(P)H), or flavin adenine nucleotide (FAD), to provide label-free optical contrast owing to biochemical or structural transformations. The sensitivity of autofluorescence measurements to changes in the molecular environment permits a robust characterization of cells and tissues, and accordingly, its potential for medical diagnosis has been intensively investigated in a number of applications [1,2,3,4,5,6,7]. While autofluorescence measurements are most commonly realized in the steady-state regime, i.e., the fluorescence intensity is measured and typically resolved in wavelength, combined multispectral and time-resolved autofluorescence measurements aim to improve the specificity of the autofluorescence readout and thus may offer a more robust approach for tissue interrogation. The potential of multispectral fluorescence lifetime spectroscopy and imaging for clinical applications has been investigated in a number of studies [8,9,10,11,12,13,14], yet it remains relatively unexploited when compared to steady-state measurements alone. This is essentially due to the complexity and cost of the instrumentation required to realize multispectral and time-resolved measurements, which typically involves multiple single-anode detectors [13], multi-anode detectors [15], monochromators [16], or other complex optical multiplexing strategies [17], ultimately making the implementation of such devices technically challenging. An additional limitation refers to the long integration times typically required to realize acquisition with a reasonable number of photons over the entire spectral range.

A common concern often associated with autofluorescence spectroscopy measurements refers to the lack of depth-resolved information in diseases where lesions not only are superficial but also extend into deeper layers of tissue. Traditionally, autofluorescence spectroscopy is carried out using excitation wavelengths in the UV to near-UV range of the spectrum [6,17,18], where endogenous fluorophores, such as collagens or NAD(P)H, are most efficiently excited [19,20]. At these wavelengths, the penetration depth of light is notoriously low (~100–200 μm [21,22]) due to the strong absorption of hemoglobin, myoglobin, lipids, or cytochrome-c [23], and single-point measurements alone cannot discern between signals originating at different depths. To overcome this limitation, autofluorescence measurements are typically realized together with a technique that can offer information from deeper layers of the tissue, such as white light diffuse reflectance [6,13,24,25], at the expense of an increased complexity that typically accompanies a multimodal setting. Recently, a fiber-based time-resolved fluorescence system was developed to interrogate the metabolic and oxygen status of internal layers of tumors in vivo [26]. While the exploration area was determined by the length of the needle with fiber rather than by excitation wavelength, this was still a single-point measurement. In alternative, autofluorescence measurements can also be realized exploiting two-photon excitation, either by means of microscopy [27,28,29] or endoscopy [30,31]. Two-photon excitation offers a larger penetration depth compared to single-photon based techniques and thus represents an attractive alternative to harness autofluorescence information from deeper layers of tissues.

In recent years, developments in complementary metal oxide semiconductor (CMOS) technology boosted the emergence of alternative detection configurations for fluorescence spectroscopy measurements. Specifically, arrays of Single Photon Avalanche Diode (SPAD) detectors have become increasingly available with integrated timing electronics that permit fluorescence lifetime measurements with ps-range temporal resolution and with unprecedented counting rate [32,33,34,35,36]. Indeed, SPAD arrays constitute attractive cost-effective alternatives to conventional strategies for multispectral fluorescence lifetime measurements [32,34], e.g., employing multi-channel photo-multiplier tubes (PMT) and time-correlated single-photon counting technology (TCSPC). The benefits of using SPAD arrays lie essentially in having integrated electronic circuitry to realize fluorescence lifetime measurements at the pixel-level rather than at the detector-level, as in conventional TCSPC, and with higher quantum efficiency compared to PMTs. This translates to relatively higher count rates that permit real-time fluorescence lifetime imaging [33,34]. 

Here, we explore the potential of SPAD arrays to realize fast and parallel time-, spectral-, and depth-resolved fluorescence measurements using fiber optics for light delivery and collection. Specifically, a fiber probe was designed to interrogate at increasing sampling depths, consisting in thirty-two collection fibers linearly arranged at increasing distances from the excitation fiber. The multispectral fluorescence signal is imaged onto a SPAD array comprising 32 × 64 pixels with independent gated counting electronics that provide temporal resolution to the fluorescence measurement. Spectral resolution is achieved by dispersing the fluorescence light across the 64 columns of the array. Light from collection fibers is imaged across the rows of the detector, thereby providing depth-resolution. We demonstrate the feasibility of this method to provide rapid multidimensional fluorescence information from fluorescent agarose phantoms and tissue samples. We believe that this method could offer a new means of tissue interrogation, particularly in clinical applications requiring the acquisition of depth-resolved information.

## 2. Materials and Methods

### 2.1. Optical Setup

The optical layout of our instrument is presented in Figure 1. The excitation light is provided by a 375 nm picosecond diode laser (BDL-SMN-375, Becker & Hickl GmbH, Berlin, Germany). A custom-designed bifurcated fiber bundle (ArtPhotonics GmbH, Berlin, Germany) is used to deliver the excitation light and to collect the corresponding fluorescence signal. The fiber bundle consists of thirty-three 100-μm core diameter fibers (NA = 0.22, length = 1.5 m), of which one fiber is used for excitation and the remaining fibers are used for collection. At the sample end, the fiber tip consists of a brass ferrule tip of 11 mm in diameter and 50 mm in length. The fibers are centered in the ferrule and arranged in line with the excitation fiber located at one end, as illustrated in Figure 1a. At the detection end, the thirty-two collection fibers are vertically oriented with the fiber that is closest to the excitation (fiber 1 in Figure 1a) on top. A 400-nm long pass filter (FEL0400, Thorlabs, Newton, NJ, USA) was added to the emission path to prevent contamination of the measurement with the excitation light. The contribution of fluorescence from the fiber to our measurements was found to be negligible. Fluorescence light emanating from the sample is dispersed by a transmission grating (GT50-06V, Thorlabs) and imaged onto a SPAD detector (SPC3, MPD, Bolzano, Italy) with a 1:1 magnification. The SPAD detector consists of 2048 pixels arranged in 32 rows by 64 columns (dimensions 4.8 × 9.6 mm). With this detection configuration, fibers that are closest to the excitation and provide information from the most superficial layers are imaged on the top rows of the SPAD array; fibers that are at a greater distance from the excitation and provide information from deepest layers are imaged towards the bottom rows of the detector. Spectral information is encoded within the 64 columns of the SPAD array. A representative intensity map of the SPAD array encoding spectral and depth information is shown in Figure 1b.

Each pixel of the SPAD detector integrates time-gating [37,38] counting electronics (in-pixel 9‑bits counter) that provide pixel-wise light intensity and temporal resolution. The pixel size is 150 µm × 150 µm, and the SPAD diameter is 30 µm, resulting in a 3.14% fill-factor. The detector features a peak Photon Detection Efficiency (PDE) of 50% at 450 nm and a median Dark Counting Rate (DCR) of 150 cps per pixel. The SPAD array can be readout at a maximum speed of 100 kfps. A complete characterization of the SPAD detector is provided in References [39,40]. In Fluorescence Lifetime Imaging (FLIM) mode, the detector outputs a 50-MHz reference clock that was used to trigger the excitation laser. The gate signal is internally generated by the SPAD camera, synchronous with the laser trigger. The gate duration and the temporal offset are user-adjustable, and they can be as short as 2 ns and 20 ps, respectively. To optimize photon collection efficiency while maintaining a reasonable temporal sampling of the fluorescence decay, we chose a sampling gate of 4 ns. For measurements of reference fluorophores, fluorescence intensity decays were sampled in 80 steps with a temporal offset of 200 ps between gates. For tissue measurements, fluorescence decays were sampled in 40 steps, with a temporal offset of 400 ps. Both these strategies resulted in an acquisition window of 16 ns, which is the maximum allowed by the SPAD electronics. The gating settings are summarized in Table 1. Exemplary measurements for each gating strategy are shown in Appendix A.

Data were transmitted to the computer via a USB 3.0 connection following each acquisition. Instrument control and data acquisition were realized via a custom software application in LabVIEW (National Instruments, Austin, TX, USA).

### 2.2. Reference Flurophores

Stock solutions of flavin adenine nucleotide (FAD, F6625, Sigma-Aldrich, Saint Louis, MO, USA) and 1,4-bis(5-phenyloxazol-2-yl) benzene (POPOP, P3754, Sigma-Aldrich) were prepared by dissolving the corresponding powder in 50 mL of purified water and ethanol, respectively, to achieve concentrations of 50 μM. We chose this particular set of fluorophores due to their distinct fluorescence characteristics when excited at 375 nm: (1) FAD fluorescence peaks at 530 nm [41] with mean fluorescence lifetimes of 2.3–3.9 ns [41,42,43,44]; (2) POPOP has a maximum emission at 420 nm [45], and its fluorescence intensity decay presents single exponential characteristics with a fluorescence lifetime of *τ* = 1.29 ns [45].

### 2.3. Fluorescence Data Analysis

Fluorescence lifetime data were analyzed using the phasor approach. A complete description and characterization of the phasor method is provided in References [46,47,48]. In brief, the phasor method is a fit-free approach that permits the robust characterization of fluorescence decays by employing the Fourier transformation of each measured decay to obtain the corresponding phasor position (g,s) in the phasor plot, according to Equations (1) and (2), respectively:(1)$$g\left(\omega \right)=\int_{0}^{T}I\left(t\right)\cdot cos\left(\omega t\right)dt/\int_{0}^{T}I\left(t\right)dt$$
(2)$$s\left(\omega \right)=\int_{0}^{T}I\left(t\right)\cdot sin\left(\omega t\right)dt/\int_{0}^{T}I\left(t\right)dt$$
where *I(t)* is the fluorescence intensity at a given time point *t* within the acquisition period *T* and *ω* is the angular frequency, given by ω=2πf, where *f* is the laser excitation frequency (i.e., 50 MHz).

Fluorescence species presenting single exponential decay characteristics are represented by a phasor that falls on the universal circle, which is defined as a semi-circle of radius 0.5 and centered at (0.5, 0). When two or more molecular species contribute to the fluorescence decay, the corresponding phasor will lie within the universal circle as a linear combination of each pure species’ phasor. Changes in the contribution of any species to the fluorescence decay will result in a shift in the phasor cloud towards the pure species’ phasor.

### 2.4. Calibration

Spectral calibration was realized by measuring the reflected signal provided by a 445-nm laser diode (Sacher Lasertechnik GmbH, Marburg, Germany) and LEDs with center wavelengths at 470 nm, 530 nm, and 630 nm (Thorlabs). The center emission wavelength of the LEDs and laser were initially measured using a microHR monochromator (Horiba, Kyoto, Japan) fitted with a Syncerity charge-coupled device (CCD) detector (Horiba) and used to calibrate our custom spectrometer. Spectral measurements of reference fluorophores FAD were compared and validated against monochromator-based measurements. A discrepancy of less than 5 nm was obtained between instruments, which is equivalent to the spectral resolution of our system. 

For time-resolved measurements, the fluorescence decay characteristics of reference fluorophores FAD and POPOP were validated against a fiber-based time-correlated single photon counting (TCSPC, SPC-730, Becker & Hickl GmbH) instrument fitted with a hybrid detector (HPM-100-40, Becker & Hickl GmbH). Measurements were realized at the emission peak of both fluorophores. The fluorescence lifetime values obtained for both fluorophores were consistent between instruments (TCSPC: *τ*_POPOP_ = 1.31 ± 0.04 ns, *τ*_FAD_ = 3.76 ± 0.05 ns; SPAD: *τ*_POPOP_ = 1.34 ± 0.06 ns, *τ*_FAD_ = 3.73 ± 0.09 ns). The instrument response function (IRF) was measured using back-reflected excitation light from reflective surfaces and by removing emission filters and grating from the optical path. The measured IRF full width at half maximum (FWHM) was 4.30 ± 0.04 ns. While the long IRF is primarily attributed to the long gates used in the fluorescence detection (4 ns), additional IRF broadening is caused by modal dispersion in the multimode fibers due to a broadening of the laser excitation pulse and corresponding fluorescence signal (approximately 200 ps).

### 2.5. Agarose Phantoms of Reference Fluorophores

In order to verify whether our system could provide depth-resolved information, we created 2 × 2 cm^2^ agarose phantoms of FAD and POPOP in various thicknesses: 1.0 mm, 1.5 mm, and 2.0 mm. The phantoms were prepared by dissolving 0.15 mg of agarose directly in 5 mL of each stock solution. Non-fluorescent water-based phantoms were also prepared. After heating, agarose solutions were poured into 3-D printed molds that were designed following the work of Mustari et al. [49]. A description of the 3-D printing process is provided in Appendix B. Fluorescence lifetime and spectral measurements were realized for each phantom and compared against the stock solutions (see Appendix A). Following the initial characterization, the phantoms were combined to create layers with different fluorescence properties, as described in Table 2 and illustrated in Figure 2a. Fluorescence measurements were realized by placing the tip of the fiber probe perpendicularly and in gentle contact with the top surface of layer 1.

### 2.6. Rabbit Aorta Specimens

Formalin-fixed rabbit aorta pieces with approximately 10 × 10 mm^2^ surface area and 2 mm in thickness were prepared as previously described [50]. Fluorescence measurements were realized by placing the fiber probe tip in gentle contact with the internal surface of the artery with the excitation light travelling towards the outer surface. Measurements were realized from different locations in multiple samples. A total of 100 measurements were realized in each location, corresponding to a total integration time of approximately 4 seconds.

## 3. Results

### 3.1. Fluorescence Measurements of Reference Fluorophores in Agarose

Figure 2b shows the fluorescence intensity maps (in log scale) for each phantom, and Figure 2c shows the corresponding normalized spectra along the rows of the SPAD detector, where data are averaged for each set of four lines of pixels. For phantom (i), we measured similar fluorescence spectra over the entire array, consistent with the measured fluorescence spectrum of FAD (see Appendix A). This resulted in an integrated spectrum that is also consistent with that of FAD (see Figure 2d, green curve). As expected, we measured higher intensities in the top rows of the array, corresponding to detection fibers closest to the excitation fiber. Particularly, the maximum intensity was measured in the third row of the array (integrated intensity ~24k photons), having decreased to 1/e of its value by row eight. 

For phantoms (ii)–(iv), we aimed to distinguish the fluorescence signals emanating from FAD and POPOP. The distance between the FAD and POPOP layers increased from phantoms (ii) to (iv) to verify the sensitivity of our system to changes in depth. In general, the results suggest that, as the distance between FAD and POPOP layers increases, the fluorescence signal from FAD becomes more dominant and the integrated fluorescence spectra approximates to that measured for phantom (i), where FAD is the only fluorophore contributing to the signal (see Figure 2d). For phantom (ii), where the FAD and POPOP layers are adjacent, the measured fluorescence spectrum changes considerably over a probed depth (see Figure 2c, black curves). While, in general, the signal is dominated by FAD fluorescence, a shoulder at ~420-430 nm consistent with a POPOP fluorescence emission becomes more prominent with depth. This shoulder becomes less evident in phantoms (iii) and (iv), where the distance between the FAD and POPOP layers increases. Particularly, in phantom (iv) (Figure 2c, blue curves), the 420–430 nm shoulder only becomes evident in the third spectral curve, corresponding to the average fluorescence spectra between rows 8 and 12. For the most top layers, the measured spectra approximate well to that of pure FAD, suggesting that the signal detected in these fibers emanates from this fluorophore only and thus from the most superficial layer of the phantom.

To quantify the contribution of POPOP to the fluorescence signal with a probed depth, we measured the ratio *I*_POPOP_/*I*_FAD_, where *I*_POPOP_ and *I*_FAD_ correspond to the average intensity measured around the fluorescence emissions peaks of POPOP (420–430 nm) and FAD (520–530 nm), respectively. The results presented in Figure 2e suggest that the contribution of POPOP is higher in phantom (ii) (black circles) and lower in phantom (iv) (blue crosses) if we exclude phantom (i). As expected, the contribution of POPOP fluorescence increases as a function of the source-detector fiber distance until stabilization is reached. Stabilization of the signal occurs approximately between rows 8–10 for phantom (ii), rows 15–17 for phantom (iii), and rows 17–19 for phantom (iv). In phantom (i), we measured a slight increase in the contribution of the 420–430 nm band (Figure 2e, green circles), which results from a general decrease in the measured fluorescence signal with depth and a corresponding decrease in the signal-to-noise ratio.

Variations in fluorescence lifetime as a function of depth were assessed via the phasor approach, and the results are presented in Figure 3. For this analysis, we only considered the top half of the detector, i.e., from rows 1 to 16, since most changes in the fluorescence signal occur in this range (as demonstrated in Figure 2) and the absolute fluorescence intensity is still sufficient to provide a robust analysis (see Figure 3b). The signals were binned over the entire spectrum and for each set of four rows, resulting in four phasor maps per phantom, which are indicative of the fluorescence decay characteristics with probed depth. In general, the results presented in Figure 3c show that the contribution of POPOP increases with probed depth, as the phasor cloud moves along the line connecting the two pure components (black dashed line) and in the direction of the pure POPOP phasor as the rows of the SPAD array increase (see Figure 3a). In contrast, as the distance between the FAD and POPOP layers increases, i.e., from phantoms (ii) to (iv), for the same group of rows, we observe a left shift in the phasor cloud towards the pure FAD component (e.g., Figure 3c, rows 1–4), which is indicative of a stronger contribution of this fluorophore to the fluorescence decay. Changes in the contribution of each fluorophore to the measured fluorescence decay were quantified by measuring the average distance of each phasor cloud with respect to the pure species’ phasors (see Figure 4d). As expected, the contribution of POPOP is stronger in phantom (ii), where the layers of FAD and POPOP are adjacent and POPOP is closer to the surface. The contribution of POPOP decreases as the distance between layers increases, i.e., from phantoms (ii) to (iv). These results are in general agreement with contributions calculated using spectral data (see Figure 2e) and also plotted in Figure 4d, for comparison.

### 3.2. Fluorescence Measurements of Rabbit Aorta

While the results obtained from agarose phantoms (Figure 2 and Figure 3) demonstrate that changes in both fluorescence spectral and lifetime distribution can be measured as a function of probed depth, the optical properties of these phantoms are far from those typically measured in biological tissues. Consequently, any conclusions derived from these results are not directly applicable to measurements of biological tissues. In order to verify the suitability of our system for measurements on biological tissues, we measured the fluorescence signal from formalin-fixed rabbit aorta specimens (see Figure 4). Figure 4a shows the characteristic lifetime phasor map calculated over the entire spectrum and depth, i.e., all photons were binned into a single channel. The phasor cloud contains small clusters of points (see arrows in Figure 4a) that originate from fluorescence decays over different spectral bands. Indeed, the phasor clouds for each spectral band (see Figure 4d, spectral bands are highlighted in Figure 4b) occupy slightly different regions in the phasor plot. For example, phasors in channels 1 and 2 are right-shifted relative to channel 3 phasors, which indicates shorter fluorescence lifetimes at shorter emission wavelengths.

With respect to depth information, we observe a consistent left shift in the phasor cloud of channel 1 with depth, towards longer fluorescence lifetimes (see Figure 4d, channel 1 panel). Interestingly, we did not observe similar shifts in other spectral bands, suggesting that changes in the fluorescence signal with depth are essentially driven by fluorophores emitting in the blue region of the spectrum. Indeed, phasor maps for the spectral bands of channels 2–4 are more compact relative to channel 1 and show little differentiation with depth. Changes in the fluorescence signal with depth are most visible in the fluorescence emission spectra (see Figure 4c). The fluorescence spectra originating from most superficial layers have maxima at around 470–480 nm. As the excitation light probes deeper layers within the tissue, the resulting fluorescence emission is shifted towards longer wavelengths, peaking at approximately 500 nm. This shift in fluorescence emission with depth is also apparent in the fluorescence intensity map, as previously shown in Figure 1b. As expected, the greatest proportion of the signal is measured in the top five rows of the array and, consequently, the total fluorescence spectrum reflects mostly the signal from this region (see Figure 4b).

## 4. Discussion and Conclusions

In this study, we explored the potential of SPAD arrays to realize parallel multidimensional fluorescence measurements. Time-, spectral-, and depth-resolved fluorescence information were retrieved from fluorescence phantoms and tissue samples. Measurements of reference fluorophores FAD and POPOP embedded in agarose demonstrated that SPAD arrays can report lifetime and spectral contrast from fluorescence signals emanating at different depths relative to the source and detector fibers (see Figure 2 and Figure 3). We note that these agarose phantoms are far from mimicking the optical properties that are typically measured in biological tissues, and thus, these measurements should be merely regarded as proof of concept. We chose this particular set of fluorescence dyes due to their strong absorption at 375 nm and their well-differentiated fluorescence spectral and lifetime characteristics that fit the purpose of this experiment.

To further demonstrate the applicability of this approach in biological specimens, we measured the fluorescence signals emanating from formalin-fixed rabbit aorta samples (see Figure 4). We note that this work aimed merely to demonstrate the technical feasibility of our approach to report time-, spectral-, and depth-resolved fluorescence information from biological specimens using SPAD arrays and fiber optics. For this purpose, rabbit aorta provided a convenient test sample due to its stratified structure and composition to demonstrate that our approach can monitor changes in the fluorescence lifetime and spectra with probed depth (see Figure 4c,d), which may be relevant in some clinical applications. In the future, we will explore the possibility of realizing clinically relevant studies where the fluorescence signatures of normal and diseased tissues can be compared in order to further demonstrate the potential and suitability of our implementation for clinical diagnostics. Despite the low penetration depth achieved with 375 nm light, as discussed below, we were still able to report lifetime and spectral contrast over different rows of the SPAD array, corresponding to a higher source-to-detector distance and, thus, to larger probed depths (see Figure 4c,d). Particularly, changes in the fluorescence signature with depth are most evident in the fluorescence spectra (see Figure 4c), where we observed a red shift of the emission peak with depth, potentially associated with changes in the scattering properties of the tissue resulting from structural and biochemical alterations over depth. Changes in fluorescence lifetime with depth are subtler and essentially confined to the short wavelength region. In the wavelength band up to 455 nm (see Figure 4d, left panel), we observed a left shift of the phasor cloud with depth, which is indicative of an increasing fluorescence lifetime. In channels 2 and 3, while the phasor clouds at different depths appear to be forming small clusters, these are closely packed together and do not suggest any specific trends in the data. From these measurements alone, it is not possible to ascertain the exact probed depth, since that would require a complete knowledge of the optical scattering and absorption coefficients of the various layers of the sample under investigation for 375 nm excitation light. Such measurements are beyond the scope of this study. Rather, our intention was merely to demonstrate the versatility of SPAD detectors and the feasibility of this approach to provide rapid multispectral depth-resolved fluorescence information in the context of single-point measurements. We note that this method could find utility in a wide range of applications from characterization of materials to clinical diagnostics.

An eventual translation of this approach to clinical measurements of biological tissues would require a priori knowledge of the tissue scattering and absorption coefficients at the excitation wavelength, so that the approximate probed depth could be directly inferred from the measurements. In biological tissue, while excitation light at 375 nm can penetrate as deep as 1.0 mm, most of the incident energy is absorbed within the first 100–200 μm, depending on the tissue type, excitation laser power, and beam width [21,22,51,52], and thus, any detected fluorescence signal is likely to originate within this region. Obviously, light penetrates deeper in tissues with a lower attenuation coefficient. For example, in blood filled thick tissues, i.e., with a high concentration of hemoglobin, fluorescence from deep layers is better detected in the red to near-infrared regions of the spectrum, given the low absorption of hemoglobin, lipids, and water in this band [23]. While the penetration depth can be generally increased by employing longer excitation wavelengths, this is achieved at the expense of autofluorescence specificity, since endogenous fluorophores such as collagens or NAD(P)H are more efficiently excited in the UV region of the spectrum [20]. To some extent, autofluorescence measurements employing two-photon excitation can obtain information from deeper layers of the tissue while also providing high lateral and axial resolution, which is key to some clinical applications. This is a considerable advantage relative to single-point approaches. We note, however, that two-photon instrumentation is considerably more complex and expensive compared to a single-point strategy employing single-photon excitation. In addition, two-photon excited autofluorescence measurements typically require long integration times to obtain multidimensional optical information, which can make this technique impractical for in vivo measurements. Moreover, the implementation with optical fibers is not trivial due to pulse broadening together with the nonlinear background signal generated within the optical fiber itself, which has a strong detrimental effect on the achievable SNR. In the context of single-point measurements for fast tissue interrogation, an implementation employing arrays of SPADs to realize simultaneously time-, spectral-, and depth-resolved fluorescence measurements would be of great added value and could raise wide interest for tissue diagnostics. One particular advantage of this method relative to similar time-resolved multispectral systems refers to the fact that lifetime, spectral, and depth fluorescence information are collected in parallel for a total acquisition time of 4 s in rabbit aorta specimens. However, we note that, as it stands, our system has a low collection efficiency and, therefore, the acquisition time could be further improved. Firstly, the fiber geometry limits the collection of photons to a single direction relative to the excitation fiber. As discussed further below, alternative fiber configurations could be explored to increase photon collection from the sample. Secondly, the time-gating architecture determines that photons collected outside the range of the temporal gate are not accounted for by the electronics and are discarded. Alternative strategies could employ parallel [53,54] or longer time-gates [55,56] for a detection of the fluorescence signal or could explore microchannel plate detectors with a position sensitive anode together with TCSPC [57]. Finally, the fill factor of each pixel is less than 3.5% [39]. Future designs could explore microlenses to focus the incident light onto the active area of each SPAD and, consequently, to increase photon efficiency [58].

Excitation light in tissue undergoes multiple scattering events prior to being absorbed and eventually generate fluorescence. Accordingly, the lateral mean propagation of photons in tissue is typically larger than the axial propagation, and the most probable migration area exhibits a banana-shaped pattern between source and detector fibers [22]. The probed depth is, therefore, a function of the source-detector distance: A larger distance implicates the detection of fluorescence from deeper layers. Previous work by Kholodtsova et al. [22] suggested that a maximum optical probed depth of 0.5 mm can be obtained in brain tissue with a source-detector separation of 1.0 mm and an excitation light at 532 nm. Our system includes thirty-two collection fibers arranged linearly at the sample end and at an incremental distance of 125 μm from the source fiber. This detection configuration would implicate that, at 532 nm excitation, a fluorescence signal from brain tissue would be detected by the first eight collection fibers only. Obviously, the range of detection would be even more limited using a 375 nm excitation light since the propagation path of photons at this wavelength would be shorter. Indeed, in our rabbit aorta experiments, over 73% of the total fluorescence signal was detected within the top five rows of the SPAD array, corresponding roughly to the first six collection fibers. While we still observed some variation of the fluorescence lifetime signal within this range (see Figure 4d, channel 1), our configuration greatly limits the depth resolution of the measurements since the fiber core diameter is of the same order of magnitude as the maximum penetration depth of the light. In order to increase the axial resolution, single-mode fibers could be employed for detection at the price of a longer acquisition time due to the poorer collection efficiency with respect to multimode fibers.

One of the characteristics of our implementation is that the spectral bandwidth for multispectral fluorescence lifetime measurements can be adjusted in post-processing. For example, additional spectral lifetime resolution could be obtained by binning the decays in eight spectral channels instead of four. We selected the wavelength bands indicated in Figure 4b since these covered the spectral emission of reference endogenous fluorophores while guaranteeing a sufficient number of photons in the fluorescence decay. The ability to fine tune the bandwidth of spectral channels for lifetime determination is an advantage relative to multispectral implementations where spectral bands are determined by the optical filters and, thus, cannot be easily modified [10,12,13,14,25,59,60,61,62]. This is relevant in cases where the fluorescence lifetime is wavelength-dependent, as is the case of our rabbit aorta measurements; see the left shift of the phasor cloud with increasing wavelength (Figure 4d).

In our implementation, fast multidimensional fluorescence measurements are implemented at the expense of spatial resolution, which constitutes a major limitation for clinical applications. While single-point approaches are commonly employed due to the simplicity of the instrumentation, the information they provide is confined to a narrow region, and multiple measurements may be necessary to provide reasonable characterization of the sample. For example, in the context of tumor identification during resection surgeries, a spatial mapping of tumor margins is essential to aid clinicians in decision making. The information provided by single-point measurements in this context would be limited. Future work will investigate alternative imaging strategies to overcome this limitation.

While our study explores the potential of SPAD detectors in the context of single-point acquisitions, we note that SPADs have been mostly exploited in fluorescence microscopy applications, such as in super-resolution [36] or measurements of protein-protein interactions via Förster resonance energy transfer (FRET) [63]. In this context, we believe that a multispectral lifetime configuration as that presented in this study could be an interesting solution for FRET applications [64,65,66] since, in principle, fluorescence lifetime measurements of the donor and acceptor could be realized in parallel over different spectral bands, thus increasing the imaging speed and spatiotemporal dynamics of the measurement.

In conclusion, in this study, we described the implementation of SPAD arrays to realize parallel time-, spectral-, and depth-resolved fluorescence spectroscopy measurements using fiber optics for excitation and collection. The feasibility of this approach was demonstrated in fluorescent agarose-based phantoms and in formalin-fixed tissue samples of rabbit aorta. Despite the low penetration of light in biological tissue, our results demonstrate that it is possible to report lifetime and spectral contrast originating at different layers within the tissue. Further work is necessary in order to employ this approach in clinical settings, namely with respect to the characterization of tissues’ optical properties and the consequent calibration for fluorescence measurements, and in the adaptation of the optical fiber design to allow greater depth resolution and photon efficiency. Overall, we believe that SPAD arrays have potential to improve the specificity of traditional fluorescence measurements and thus can find wide utility in clinical applications for tissue interrogation and surgical guidance using fiber optic devices.

## Figures and Tables

**Figure 1 sensors-19-02678-f001:**
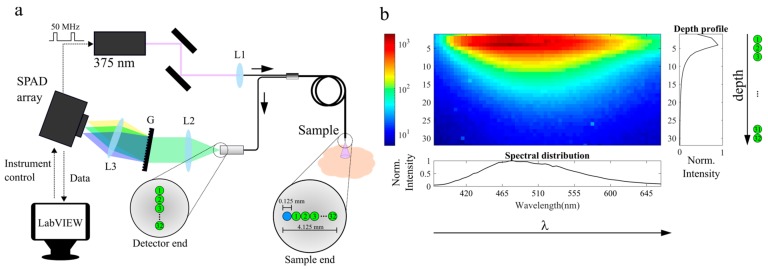
(**a**) The optical layout of the instrument and fiber arrangement at the sample and detection ends and the electronic chain for instrument control, synchronization, and data acquisition. The dashed arrows indicate electronic connections. The excitation fiber is highlighted in blue; the detection fibers are numbered and highlighted in green. Numeration of the fibers increases with the distance to the excitation fiber. At the detection end, the fibers are arranged vertically and perpendicularly to the long axis of the Single Photon Avalanche Diode (SPAD) array. Detection fibers are imaged with a 1:1 magnification onto the SPAD detector. The distance to excitation fiber increases from top to bottom as indicated by the numeration. Fluorescence light is dispersed and spectrally resolved across the long axis of the detector using a transmission grating comprising 600 grooves/mm. L1: Aspheric lens, f = 11.0 mm (A220TM, Thorlabs); L2 and L3: Bi-Convex Lens, f = 60.0 mm (LB1723-B, Thorlabs). (**b**) An intensity map measured in the SPAD array for a representative fluorescence measurement of formalin-fixed rabbit aorta: The fluorescence intensity variation with depth and wavelength are plotted on the right and bottom, respectively.

**Figure 2 sensors-19-02678-f002:**
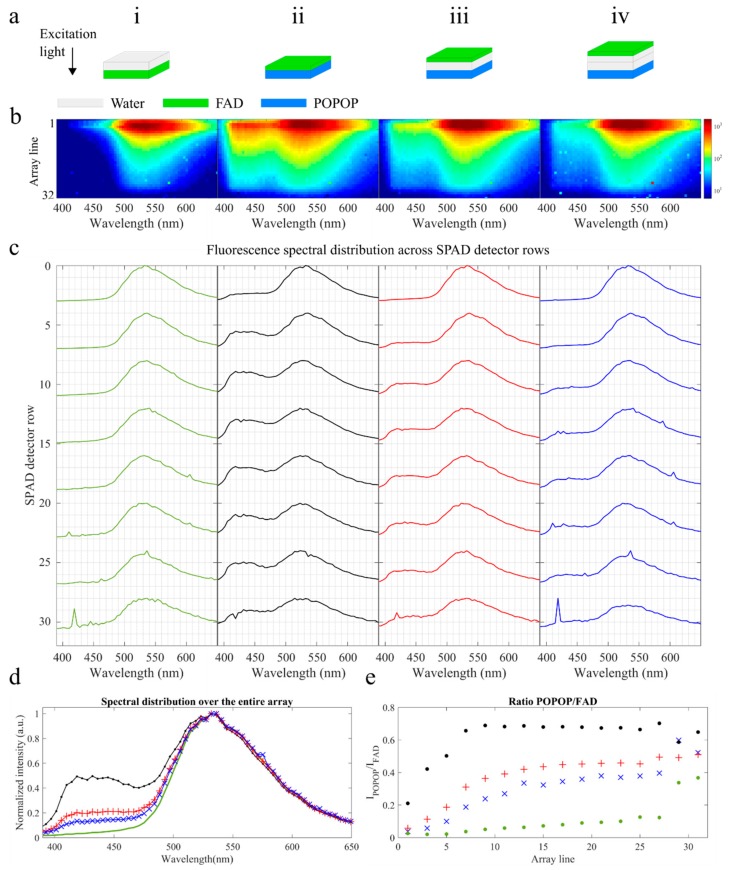
Spectral distribution of the fluorescence signal with the distance from the excitation fiber measured in agarose phantoms of flavin adenine nucleotide (FAD) and 1,4-bis(5-phenyloxazol-2-yl) benzene (POPOP): (**a**) A diagram of the phantoms as described in Table 2: The black arrow indicates the direction of excitation light. (**b**) Intensity maps measured for the phantoms indicated in (i)–(iv). (**c**) Normalized intensity profiles per line in the SPAD array: The profiles are binned for each set of four lines of pixels. The curves on the top represent fluorescence signals collected by fibers closer to the excitation fiber. The sharp spurious peaks visible in the fluorescence spectra (e.g., phantoms (i) and (iv), bottom rows, 420 nm) are artefacts originating from data normalization and dark count subtraction. (**d**) Fluorescence emission spectra integrated over all rows of the SPAD detector for each phantom: The color of each curve corresponds to the colors in Figure 2c. (**e**) Intensity ratio calculated over the spectral bands 8–12 and 31–35 pixels, indicating the ratio of POPOP over that of FAD for each line of the array: Ratios closer to 1 indicate a higher contribution of POPOP to the fluorescence signal. The color legend for the phantoms are (i) green solid line; (ii) black circles; (iii) red plus sign; and (iv) blue crosses.

**Figure 3 sensors-19-02678-f003:**
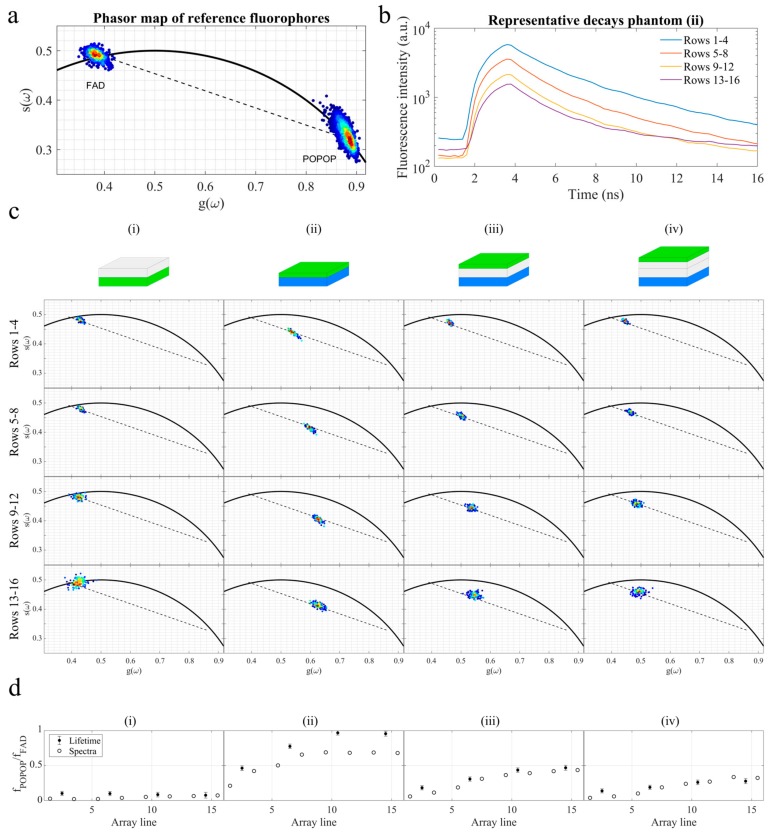
(**a**) A fluorescence lifetime phasor map of FAD and POPOP reference solutions (n = 100 measurements per solution). (**b**) Representative fluorescence intensity decays of phantom (ii) for each group of rows. (**c**) Phasor maps of FAD and POPOP phantoms (n = 100). Columns (i) to (iv) indicate phasor maps of corresponding phantoms as illustrated in Figure 2 and reported on top of each column. Each phasor map shows the phasor transformation of fluorescence decays binned for the rows indicated on the left and for a single spectral channel comprising all columns of the SPAD detector. The black dashed lines connect the center of FAD and POPOP phasor clouds obtained from reference solutions; see Figure 3a. (**d**) The ratio of POPOP to FAD for the top 16 lines of the SPAD array, calculated via lifetime phasors (filled circles) and fluorescence spectral data (empty circles). For lifetime phasors, data were binned for each 4 rows; for spectral data, data were binned for each 2 rows. Ratios closer to 1 indicate a higher contribution of POPOP to the fluorescence signal.

**Figure 4 sensors-19-02678-f004:**
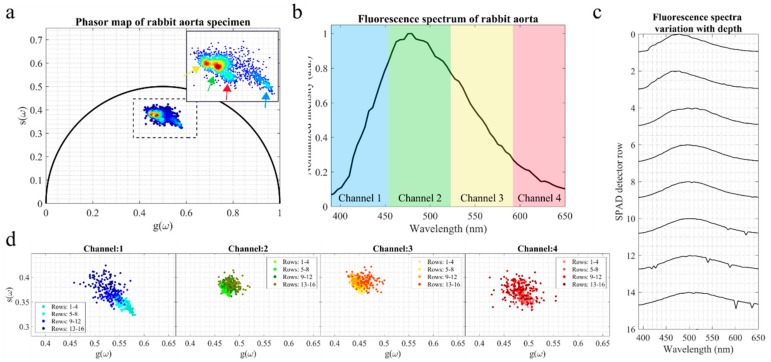
Fluorescence measurements of fixed rabbit aorta for the top 16 rows of the SPAD detector (n = 100 measurements). (**a**) A lifetime phasor map for the entire array, i.e., fluorescence decays were binned over the entire spectrum and the top 16 rows of the detector. The panel on the top right shows the zoomed phasor map of the area enclose by the black dashed rectangle. The arrows indicate clusters of phasors that were formed from data obtained at different wavelengths. The color of each arrow represents a spectra channel as indicated in Figure 4b. (**b**) Integrated fluorescence spectrum. (**c**) Variation of fluorescence spectra with probed depth. (**d**) Fluorescence lifetime phasor maps for spectral channels identified in Figure 4b and within the region demarcated by the dashed rectangle in Figure 4a. For each spectral channel, fluorescence decays were binned by distance from the excitation fiber, which is indicated by the color gradient of the phasor. Lighter colors represent data from collection fibers closer to the excitation and thus emanating from the most superficial layers of the tissue.

**Table 1 sensors-19-02678-t001:** The time-gating strategy for fluorescence lifetime measurements of fluorescence standards and biological tissue.

	Reference Fluorophores	Tissue Measurements
Gate width	4 ns	4 ns
Gate shift	200 ps	400 ps
Number of gates	80	40

**Table 2 sensors-19-02678-t002:** Reference fluorophores and thickness of each layer of the phantoms studied: The tip of the fiber probe was placed in contact with the top surface of layer 1. The distance to the fiber probe increased from layers 1 to 4. A diagram of each phantom is shown in Figure 2a.

Phantom	i	ii	iii	iv
Layer 1	Water (2.0 mm)	FAD (1.0 mm)	FAD (1.0 mm)	FAD (1.0 mm)
Layer 2	FAD (2.0 mm)	POPOP (2.0 mm)	Water (2.0 mm)	Water (1.5 mm)
Layer 3	-	-	POPOP (2.0 mm)	Water (2.0 mm)
Layer 4	-	-	-	POPOP (2.0 mm)

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
