# Peer review of "Multispectral Depth-Resolved Fluorescence Lifetime Spectroscopy Using SPAD Array Detectors and Fiber Probes"

_sensors, 2019, doi:10.3390/s19122678_

Round 1
Reviewer 1 Report
In this paper, the authors built a multispectral depth-resolved fluorescence lifetime spectroscopy with SPAD array. The performance has been shown in proof-of-principle with agarose phantoms of two reference fluorophores and show clearly the spectrum variation when the distance between the two fluorophores varies. They also tried the spectroscopy on real biological samples on rabbit aorta.
It has enough novelty in setting up the spectroscopy. However, for the biological sample measurement, the spectrum actually didn’t give much information. For the spectroscopy to be applicable as diagnostics, control samples and diseased samples should be compared side by side. Clear and consistent changes in the spectrum would give instructions to the doctors that something wrong with an unknown sample. The current experiment on the rabbit aorta only worked as control. I would suggest the authors to measure some diseased samples with atherosclerosis and compared with the healthy control to see if there is any distinguishable variation.
Other minor comments:
In Fig. 1c, there are two clear peaks at 420nm for phantom (i) and (iv) in row 30, which is the emission peak of POPOP. However, for phantom (i), there is no POPOP in the sample. But the two peaks are too high to be considered as noise. Any possible reasons causing the peak at 420nm?
Author Response
please see file attached

Reviewer 2 Report
Referee report on the paper sensors-524758 for publication in Sensors
The paper is dedicated to the use of SPAD arrays in biochemical and biomedical
research exploring the application of SPADs together with
fiber-optic based delivery and collection geometry.
In my opinion the paper is very interesting, the only revision I require is to improve information on the setup adding informations about the adopted electronic chain. Moreover I think that would be useful to improve sections 2.2, 2.3, 2.4 adding images and more informations.
Reviewer 3 Report
The authors report a time-gated phasor approach for the imaging of endogenous fluorophores with potential clinical application in tissue diagnostic. Indeed, a non-invasive imaging of metabolic activities of naturally fluorescent materials has undeniable value for metabolic studies, but even further utility can be found in clinical applications for fast tissue interrogation during surgical resection of tumors.
The entire paper is focused on the possibility of configuring SPAD for time-, spectral- and depth-resolved fluorescence spectroscopy measurements, therefore a systematic comparison with conventional fiber-based TCSPC systems is provided, and an evaluation of the accuracy of measurements is presented by a widely used fitting- free approach.
Authors mention that depth-resolved information could be directly obtained from the measurements of reference fluorophores in phantoms, but despite that, the penetration and detection efficiency in tissue is not clearly compared/discussed regarding to 2-photon approaches.
Author Response
please see file attached

Reviewer 4 Report
The authors describe very interesting work based on SPAD array detectors for spectral fluorescence lifetime measurements. They have designed an ingenious experimental set-up based on a multimode optical fibre bundle that allows the spectral and lifetime information to be collected from different depths of the sample. They use 375nm excitation, and employ FAD and POPOP phantoms to demonstrate their method, and then apply it to a rabbit aorta. The idea is great, and the data collected very persuasive. Maybe the data analysis could be refined in the future – I think there may well be more information lurking in it than currently shown.
The manuscript is very well-structured and written; it is concise and straight-forward to follow, with a clear introduction and motivation, putting the work into the relevant context. The work is of interest to the fluorescence spectroscopy and microscopy community, and to researchers working on tissue diagnostics, perhaps in a clinical setting.
I have some minor comments that the authors may wish to address before publication:
1) In the first line of the abstract “Single Photon Avalanche Diode (SPAD) arrays have been increasingly exploited…” maybe say “…are increasingly exploited….”
2) p2, line 66. Another way to get to deeper layers is with two-photon excitation, and commercial instruments (jenlab) are used to do this in clinical applications, see
Dancik, Y., Favre, A., Loy, C.J., Zvyagin, A.V., Roberts, M.S., 2013.Use of multiphoton tomography and fluorescence lifetime imaging to investigate skin pigmentation in vivo. J. Biomed. Opt. 18, 26022.
Sanchez, W.Y., Prow, T.W., Sanchez, W.H., Grice, J.E., Roberts, M.S., 2010. Analysis of the metabolic deterioration of ex vivo skin from ischemic necrosis through the imaging of intracellular NAD(P)H by multiphoton tomography and fluorescence lifetime imaging microscopy. J. Biomed. Opt. 15, 046008.
This would be worth a brief discussion or mention. The jenlab webpage also has information on this. (This comment also applies to p10, line 358.)
3) p4, table 1. Do the gate width and shape depend on the count rate? Some of the SPAD arrays with gating (swiss spad?) have such a feature, I seem to remember.
4) p5, line 180. Why is the IRF so long? 4.3ns? Is this due to optical broadening in the multimode fibres, or is the SPAD time resolution responsible for this? Or a combination of those two?
5) p7, line 235 “contiguous” layers. Do the authors mean “adjacent”, or “next to each other”?
6) p8, fig 3. Maybe show a representative fluorescence decay, so the raw data quality can be assessed by the reader.
7) p8, fig 3, data analysis. Would it be possible to do double-exponential fits to the data, with the pre-exponential factors representing the spectra of the two dyes? Or in the phasor representation, would it be possible to plot the distance of the data cloud along the line for each spectral window? This should be related to the spectrum.
8) p8, line 283, rabbit aorta. Do the authors have a picture of it?
9) p9, figure “1” should be figure “4”
10) p11, line 399. It’s ok to bin, but is the noise then not binned too? Would it be better to engineer the optics such that the desired spectral band falls onto one SPAD?
11) refs 13, 18, 22, 26, 28, 29, 31, 37 (also volume number), 49 (also volume number), 51, 52, 53, 54, 59, 61, 62 are missing full page numbers, refs 14, 16 journal details. Ref 50, author Neil should have capital initials (M.A.A.), ref 54 author Jo also (Jo, J.A.)
Addressing some of these points would increase the clarity and maybe impact of the manuscript.
Author Response
please see file attached

Round 2
Reviewer 1 Report
Now I think the paper is in a good shape for publication.